# Discovery of Nitro-azolo[1,5-*a*]pyrimidines with Anti-Inflammatory and Protective Activity against LPS-Induced Acute Lung Injury

**DOI:** 10.3390/ph15050537

**Published:** 2022-04-27

**Authors:** Alexander Spasov, Vadim Kosolapov, Denis Babkov, Vladlen Klochkov, Elena Sokolova, Mikhail Miroshnikov, Alexander Borisov, Yulia Velikorodnaya, Alexey Smirnov, Konstantin Savateev, Victor Fedotov, Svetlana Kotovskaya, Vladimir Rusinov

**Affiliations:** 1Department of Pharmacology & Bioinformatics, Scientific Center for Innovative Drugs, Volgograd State Medical University, Volgograd 400131, Russia; aspasov@mail.ru (A.S.); vad-ak@mail.ru (V.K.); klochkovvladlen@gmail.com (V.K.); sokolova210795@gmail.com (E.S.); mirwaisroman77@gmail.com (M.M.); borissow1978@rambler.ru (A.B.); alta-u@mail.ru (Y.V.); alexeysmirnov.volggmu@gmail.com (A.S.); 2Department of Organic and Biomolecular Chemistry, Ural Federal University Named after the First President of Russia B.N. Yeltsin, Mira Street, 19, Yekaterinburg 620002, Russia; i-krafttt@yandex.ru (K.S.); vicww4@gmail.com (V.F.); sk-kotovskaya-665@yandex.ru (S.K.); v.l.rusinov@urfu.ru (V.R.)

**Keywords:** cytokine, IL-6, inflammation, acute lung injury, macrophage, azolo[1,5-*a*]pyrimidines

## Abstract

Acute lung injury remains a challenging clinical condition, necessitating the development of novel, safe and efficient treatments. The prevention of macrophage M1-polarization is a viable venue to tackle excessive inflammation. We performed a phenotypic screening campaign to identify azolopyrimidine compounds that effectively inhibit LPS-induced NO synthesis and interleukin 6 (IL-6) secretion. We identified lead compound **9g** that inhibits IL-6 secretion with IC_50_ of 3.72 µM without apparent cytotoxicity and with minimal suppression of macrophage phagocytosis in contrast to dexamethasone. In a mouse model of LPS-induced acute lung injury, 30 mg/kg i.p. **9g** ameliorated anxiety-like behavior, inhibited IL-6 release, and limited neutrophil infiltration and pulmonary edema. A histological study confirmed the protective activity of **9g**. Treatment with compound **9g** prevented the migration of CD68^+^ macrophages and the incidence of hemorrhage. Hence, we have identified a promising pharmacological approach for the treatment of acute lung injury that may hold promise for the development of novel drugs against cytokine-mediated complications of bacterial and viral infections.

## 1. Introduction

Acute lung injury (ALI) during bacterial and viral infections remains a major healthcare problem, which was further highlighted by the current COVID-19 pandemic. According to recent multicenter cohort studies, ALI is characterized by 34–58% mortality rates, and 64–77% of the patients affected develop acute respiratory distress syndrome [1]. Available treatments mainly include systemic glucocorticoids and monoclonal antibodies against interleukin receptors to suppress M1-macrophage polarization in a vicious cycle of cytokine release syndrome. Methylprednisolone attenuates ALI in rodent models [2,3] and reportedly improves clinical outcomes in COVID-19 patients [4,5,6]. However, the use of glucocorticoids is limited due to potential complications—compromised innate immunity, glycemic variability, and the impairment of endogenous cortisol production [7]. Early treatment with recombinant IL-1 receptor antagonist anakinra was shown to reduce mortality rates in severe COVID-19 pneumonia [8,9]. At the same time, the IL-6 receptor antagonist tocilizumab failed to show apparent benefits [10]. In addition, monoclonal antibody therapy has limited availability in low and middle-income countries. Hence, the development of novel, specific and safe therapies for ALI is a priority task.

Viral infections, including COVID-19, may result in excessive activation of tissue macrophages and neutrophil recruitment, resulting in overproduction of inflammatory factors, diffuse alveolar damage, and lung edema [11]. Viral nucleic acids activate endosome TLR3/7 receptors. The activation of TLR triggers intracellular signaling through the transcription factor NFκB and AP-1 to induce gene transcription of pro-inflammatory cytokines and inducible nitric oxide synthase (iNOS). As a result, the release of cytokines and nitric oxide damage lung cells, increases vascular permeability and attracts circulating immune cells, further exacerbating tissue damage.

Fused pyrimidines have a long history of medicinal chemistry applications. Being ATP-mimetics, they are especially useful as a scaffold for the development of kinase inhibitors [12]. A range of pyrazolopyrimidine derivatives are known to inhibit kinases involved in inflammatory and autoimmune diseases, e.g., JAK3 (**1**) [13], BTK (**2**) [14], and IRAK4 (**3**) [15], or specifically protect from LPS-induced ALI (**4**) [16] (Figure 1). A pyrazolopyrimidine core with lipophilic *N*-substituent featuring additional H-bond acceptors is a common scaffold for the class.

Previously, we have demonstrated that nitro group plays an essential role in the manifestation of useful biological activity of azoloazine heterocycles [17] (Figure 2). We have shown that this class possesses a wide range of antiviral actions [18,19] and an affinity toward adenosine receptors [20] that is associated with improved survival in LPS-induced septic shock [21]. Furthermore, some derivatives of this heterocycle series demonstrated higher antiglycation activity than a reference compound, aminoguanidine, and could be considered candidates for extended studies to produce drugs against complications of diabetes mellitus [22,23]. Additionally, it was shown that nitro-derivatives of this series demonstrate low toxicity [17], which is uncommon for nitro-compounds. Moreover, the nitro group is a convenient moiety for further modifications, thus making it possible to synthesize potent biologically active compounds, such as azolopurines. Additionally, the nitro group enhances NH-acidity in azolopyrimidines that permits the formation of water-soluble salts to facilitate biological evaluation.

Hence, the purpose of the present work was to evaluate novel members of the nitroazoloazine class, such as nitrotetrazolopyrimidines, nitrotriazolopyrimidines and water-soluble dinitroderivatives, as potential agents against excessive inflammation during LPS-induced lung injury.

## 2. Results

### 2.1. Chemistry

The synthesis of target nitro derivatives of the azolo[1,5-*a*]pyrimidine series **9** and **10** was performed via chlorodesoxygenation of nitroazolopyrimidones **5** and **6** with the phosphoryl trichloride–pyridine–acetonitrile system, which was developed by us earlier [24]. This reaction medium enabled us to isolate intermediate halogen derivatives **7** and **8**, which were subsequently converted into heterocycles **9** and **10** by nucleophilic substitution with the corresponding alkylamines (Figure 1).

The obtained alkylaminotetrazolopyrimidines **9a**–**g** are slowly crystallizing solids with low melting points due to their azide-tetrazole tautomerism. The occurrence of different tautomeric forms was confirmed by doubled signals with the same multiplicity in ^1^H NMR spectra of the obtained products. The largest difference in chemical shifts was observed for NH proton: thus, two broadened multiplets were observed in the spectrum of isopropyl derivative **9a** at 5.58 and 8.19 ppm with a 1:0.33 ratio of the integrals. The elemental analysis confirmed the proposed molecular formula and also led to a conclusion that the aforementioned ^1^H NMR signals belong to the tautomeric forms of heterocycle **9a**. At the same time, the proton signals of C(5)–CH_3_ and CH(CH_3_)_2_ groups did not exhibit different chemical shifts for individual tautomeric forms, but the single proton multiplets of the CH group overlapped in the spectrum with a difference of merely 0.1 ppm. A similar situation was also observed for **9b**, **9d,** and **9e** heterocycles, as there were two sets of signals accordingly to azido-tetrazole tautomerism. On the other hand, compounds **9c**, **9f,** and **9g** gave only one set of ^1^H NMR signals, indicating the presence of only one tautomer. IR spectra of the obtained heterocycles **9a**–**g** contained absorption bands in the regions of 1320 and 1590 cm^−1^, corresponding to the vibrations of nitro groups, as well as a weak absorption band in the region of 3350 cm^−1^ due to the NH bond vibrations. On the other hand, all IR spectra of the obtained 7-alkylamino-6-nitrotetrazolopyrimidines **9a**-**g** featured a strong absorption band in the region of 2130 cm^−1^, pointing to the existence of the indicated heterocycles in azide form when isolated in the crystalline state.

Compounds **10f** and **10g** were crystalline solids with R_f_ 0.5–0.8 in EtOAc in thin-layer chromatography, while the starting pyrimidone **6** has R_f_ 0.0 in this eluent. We observed characteristic signals in the ^1^H NMR spectra of azolopyrimidines **10f** and **10g** corresponding to propargyl groups: triplets in the region of ~2.3 ppm and doublets in the region of ~4.0 ppm.

An alternative synthetic route was applied for the synthesis of water-soluble salts of dinitrotriazolo[1,5-*a*]pyrimidines **13** and **14**. It includes nitration of the azoloazine **11**, which leads to electrophilic substitution of hydrogen with nitro-group in both the pyrimidine cycle and the furyl substituent to form a dinitro derivative **12**. Heterocycle **12** is a strong NH-acid due to the electron-withdrawing effect of the nitro group. This property was used to synthesize water-soluble forms of nitroazolo[1,5-*a*]pyrimidines **13** and **14** by reaction with aminoguanidine bicarbonate and guanidine hydrochloride (Figure 2).

The purity and structure of all heterocycles were established by IR, 1H and 13C NMR, and elemental analysis (see Appendix A).

### 2.2. Effect of Compounds on Pro-Inflammatory Activation of Macrophages

Once in hand, we evaluated the effect of target compounds on the pro-inflammatory activation of murine peritoneal macrophages at a fixed concentration of 100 μM as an accumulation of nitric oxide in the cell supernatants. The results are presented in Table 1. Structure–activity analysis shows that the anti-inflammatory activity depends on the position and nature of the substituents in 5-methyl-6-nitro-7-akylaminetetrazolo[1,5-*a*]pyrimidine. Thus, an alkyl substituent at the *N*^7^ enhances activity, especially in the case of 2-phenylethyl derivatives **9f** and **9g**. In the case of 2-propargylthiotriazole[1,5-*a*]pyrimidines **10f** and **10g,** activity drops sharply. These compounds also turned out to be cytotoxic, possibly due to the covalent modification of cellular proteins. The activity of water-soluble forms of azolo[1,5-*a*]pyrimidines was found to be dependent on the nature of the cation since guanidine salt **14** did not demonstrate inhibition of nitric oxide synthesis, while the activity of aminoguanidine derivative **13** was comparable to that of the tetrazolopyrimidine series. Noteworthy, aminoguanidine is known as a selective inhibitor of iNOS itself [27], which apparently explains the activity of compound **13**. Hence, it was excluded from further study.

The most active compound, **9g**, comprising *N*^7^-2-(4-hydroxyphenyl)ethyl moiety, was further studied in a wide concentration range to determine the effect on nitric oxide synthesis, interleukin 6 (IL-6) secretion, and macrophage viability (Figure 3). Compound **9g** inhibited both NO synthesis and IL-6 synthesis with IC_50_ of 3.72 μM. The compound did not affect the viability of the macrophages up to a concentration of 50 μM.

### 2.3. Effect of Compound **9g** on Macrophage Phagocytosis

Next, we evaluated lead compound **9g** for immunodepressant activity in parallel with the reference drug dexamethasone. Yeast phagocytosis was assessed by microscopic examination as the number of phagocytes per 100 macrophages and as the number of phagocytosed yeast cells in each phagocyte. Cell viability was assessed as lactate dehydrogenase activity in cell supernatants, which correlates with increased cell membrane permeability.

After 24-h of incubation, dexamethasone, but not compound **9g**, significantly suppressed phagocytosis (data not shown). As shown in Figure 4, after 72 h of incubation, the proportion of phagocytes in the dexamethasone-treated cells further decreased from 57% to 41%, which coincides with the literature’s data [28,29]. The mean number of phagocytosed yeast cells was decreased by 59%. Compound **9g** preserved macrophage phagocytic activity and reduced the mean number of phagocytosed particles by 16%. No cytotoxic properties were noted for the tested compound during the experiment. In addition, macrophage spreading was visually scored. Spreading is a well-known phenotypic marker of cell activation, and phagocytic capacity is limited by the amount of available membrane [30,31]. Yeast non-significantly stimulated macrophage spreading, while **9g** treatment reduced it to a nearly unstimulated level. The effect of dexamethasone was even more significant. We may conclude that dexamethasone showed a pronounced immunosuppressive effect, while compound **9g** preserved the innate phagocytic activity of macrophages.

### 2.4. Protective Activity of **9g** in LPS-Induced Acute Lung Injury

Given the promising cellular data, we studied the protective activity of lead compound **9g** in a model of LPS-induced acute lung injury. The randomization of C57bl/6j mice into experimental and control groups was based on body weight and behavioral activity assessed in an open field test 24 h before the start of the experiment. The next day, mice were treated with vehicle or test compounds 1 h before intratracheal instillation of LPS solution in sterile saline.

The behavioral activity of animals was followed 2 and 24 h after LPS administration with an open field test. LPS-treated mice showed a pronounced decrease in motor activity accompanied by lethargy and sedation (Figure 5). Horizontal motor activity at the 2-h point demonstrated a significant decrease relative to intact control that received saline only. Motor alterations persisted for up to 24 h. For mice from the LPS + dexamethasone group, a decrease in horizontal activity was observed after 2 h, comparable to that of the LPS group, but at the 24-h point, a significant improvement was evident. Compound **9g** also restored motor activity of animals comparably to dexamethasone.

We observed similar trends in exploratory behavior as well. LPS-treated mice avoided the center of the field, while both dexamethasone and **9g** restored the number of center entries and time in the center to intact levels.

Subsequently, we assessed the degree of inflammation and lung injury using a range of biochemical, cytological, and morphological parameters. As shown in Figure 6, the LPS administration led to the secretion of pro-inflammatory IL-6 in the bronchoalveolar lavage (BAL) and blood plasma, while TNF-α in plasma was even reduced compared to initial values. The development of acute lung damage is confirmed by an increase in the permeability index of alveolar vessels for blood plasma proteins. Dexamethasone and compound **9g** normalized these markers of inflammation and lung damage. In the case of IL-6 blood plasma, a statistically significant difference was observed. Thus, we have shown that dexamethasone and **9g** prevent the increase in permeability of pulmonary vessels and counteract the development of pulmonary edema.

Upon examination of the cellular content of the BAL, a sharp increase in the content of mature segmented neutrophils was observed compared to a group of intact animals (*p* < 0.05) accompanied by a sharp drop in the content of monocytes, which indicates the development of an acute inflammatory process (Figure 7). Conversely, dexamethasone and **9g** treatment significantly decreased the segmented neutrophils count and increased the content of monocytes in BAL. At the same time, **9g** had a less pronounced effect on segmented neutrophils migration, i.e., 37.6% of BAL cells were neutrophils, while for dexamethasone, their number was 22.5%, which is much closer to the intact value.

The leukocyte count in control animals with LPS-induced inflammation showed a similar pattern. Dexamethasone has a pronounced ability to ameliorate inflammatory processes, significantly reducing the proportion of mature segmented neutrophils compared to those in animals from the LPS group. The change in the leukocyte formula after **9g** treatment is also due to the restoration of the neutrophils to lymphocytes ratio. Both dexamethasone and **9g** normalized leukocyte subpopulations to the values of intact animals.

Spleen mass and lymphocyte content were also assessed as markers of immunodepression. The difference for both markers was nonsignificant between intact, LPS-, or LPS+ **9g**-treated mice, while dexamethasone significantly reduced them.

Histological examination of the lung 24 h after LPS administration revealed massive infiltration with polymorphonuclear neutrophils in the interstitial tissue and purulent exudate in the lumen of the alveoli and pulmonary bronchioles (Figure 8) compared with animals of the control group without LPS. Immune cell infiltration was associated with diffuse alveolar damage and interstitial edema by congestive atelectasis. In the case of local swelling of alveolar walls, compensatory thinning and overstretching of interalveolar septa were noted in another part, accompanied by diapedesis of erythrocytes. After administration of dexamethasone to LPS-treated animals, moderate thickening of the interalveolar septa was observed due to edema and the accumulation of macrophages and macrophage-like cells in the interstitial tissue. The lung tissue of animals administered with **9g** was characterized by limited focal infiltration of the interalveolar septa with neutrophils.

To specifically elucidate immune cell infiltration, we used immunostaining for CD68 glycoprotein, which is a specific marker for monocytes and macrophages (Figure 9) [32]. In the lung tissue of intact animals, CD68^+^ cells were localized in the adventitia of the respiratory bronchioles. Further, individual immunopositive cells were found on the alveolar surface, in the interalveolar septa, and in the interstitium of the lungs. LPS administration led to a massive accumulation of CD68^+^ macrophages in the thickened walls and on the surface of alveoli, as well as among infiltrating cells. In contrast to LPS, for LPS + dexamethasone treatment, large CD68^+^ cells, i.e., macrophages, were rarely detected. In addition, single small CD68^+^ cells were located on the surface of alveolar walls. Treatment with **9g** also decreased the immunoreactivity of the lung tissue, and the cellular composition of the inflammatory infiltrate was represented mainly by mononuclear cells and CD68^+^ macrophages, which were located mainly in the thickened interalveolar septa and less often on the surface of the alveoli.

We also performed a semi-quantitative assessment of morphological markers of inflammation as described previously [33]. Lung tissue damage was determined as the incidence of hemorrhage and neutrophil infiltration. It was shown that LPS-induced lesions were significantly ameliorated by dexamethasone and compound **9g** with comparable efficacy (Figure 10).

## 3. Discussion

Pulmonary macrophages play important protective roles in the lungs, maintaining innate immunity in the respiratory system. At least two populations of macrophages are recognized in the healthy lung: alveolar macrophages, freely localized in the airways, and resident or interstitial macrophages, located in the lung parenchyma [34]. Lung macrophages are long-lived cells, and their repopulation occurs due to in situ proliferation but not from the bone marrow [35,36].

During airway or systemic infection, PAMPs or DAMPs are being recognized by macrophage TLRs to trigger complex signaling pathways to ultimately activate genes of inflammatory response. Downstream kinases include, but are not limited to, JAK1/3, Tyk2, IRAK1/4, BTK, GSK3B, etc. They are considered promising and validated targets for the treatment of inflammatory and autoimmune disorders, such as rheumatoid arthritis, inflammatory bowel disease, Crohn’s disease, and many others, including sepsis, which was reviewed recently [37,38]. However, it is unclear which target holds the most promise, and the development of highly selective kinase inhibitors remains a challenging task.

Therefore, we choose the phenotypic drug discovery approach to identify novel lead compounds against cytokine overproduction and inflammation-induced tissue damage. In this case, we can avoid failures associated with in vivo validation of hits and leads identified in biochemical target-based screening. Instead, phenotypic leads were naturally selected in a cell-based model. Primary mouse macrophages were stimulated by *E. coli* LPS to induce M1-polarization. Firstly, the activity of the compounds was evaluated as the ability to inhibit NO synthesis, and compounds that suppress it by 40% or more at a concentration of 100 μM were considered promising if their cytotoxicity was less than 20%. In the second step, hit compounds were studied in a wide range of concentrations as inhibitors of NO synthesis and IL-6 secretion, which is considered a key mediator of cytokine release syndrome and severe COVID-19 [39,40,41,42]. A similar approach has been previously used by other authors to identify anti-inflammatory compounds [16,43] and proved to be successful.

To sum up, the biological evaluation of the nitroazolo[1,5-*a*]pyrimidine series revealed a lead compound **9g**, a micromolar inhibitor of NO synthesis and IL-6 secretion in LPS-stimulated C57BL/6J peritoneal macrophages. Low cytotoxicity of the compound (CC_50_ >100 μM) was confirmed with both MTT and LDH tests. Importantly, while dexamethasone has superior anti-inflammatory potency, compound **9g,** in contrast, exerts a negligible effect on the phagocytic activity of macrophages, which renders it a safer alternative for long-term treatment, especially in the case of concomitant infections. Hence, the phenotypic approach allowed us to identify lead compound **9g** that is simultaneously cell-permeable, non-cytotoxic, prevents IL-6 over-secretion, and produces minimal immunodepressive activity.

In a mouse model of LPS-induced acute lung injury, compound **9g** restored anxiety-like behaviors caused by LPS, reflected in the amelioration of motor and exploratory activity in an open field test [44]. Dexamethasone and **9g** also attenuated LPS-induced hypothermia in mice. A similar effect was observed for dexamethasone previously [45].

The positive effect of dexamethasone and **9g** was also evident in biochemical and cytological markers. Both dexamethasone and **9g** normalized the blood plasma IL-6 level and ameliorated alveolar vascular permeability. Of note, the secretion of TNF-α a day after the LPS administration was even reduced in control mice, which is consistent with data of other authors [46,47]. Treatment with dexamethasone or **9g** reduced the number of neutrophils in blood and bronchoalveolar lavage fluid. The protective activity of **9g** was confirmed with a morphological examination of lung tissue. While LPS alone produced massive immune cell infiltration, hemorrhage, and pulmonary edema, compound **9g** significantly reduced tissue lesions and the migration of CD68^+^ cells.

Possible limitations of the study should be mentioned as well. First, a limited number of evaluated compounds does not allow to reliably deduce structure–activity relationships; hence more potent compounds may be overlooked. Second, similarly to other phenotypic drug discovery projects, the biological target of **9g** remains unknown. Target deconvolution is a tedious task, but it is necessary for follow-up structure-based optimization and assessment of on-target toxicity. These questions are to be investigated in our future studies.

## 4. Materials and Methods

### 4.1. Synthesis

Commercial reagents were obtained from Sigma-Aldrich (St. Louis, MO, USA), Acros Organics (Geel, Belgium), or Alfa Aesar (Ward Hill, MA, USA) and used without any further purification. All workup and purification procedures were carried out using analytical-grade solvents. One-dimensional ^1^H and ^13^C NMR spectra were acquired on a Bruker DRX-400 instrument (400 and 101 MHz, respectively), utilizing CDCl_3_ and DMSO-*d*_6_ as the solvent and as an external reference. The following abbreviations are used for a multiplicity of NMR signals: s—singlet, d—doublet, t—triplet, q—quartet, dd—double of doublets, m—multiplet, br—broaded. IR spectra were recorded on a Bruker Alpha spectrometer equipped with a ZnSe ATR accessory. Elemental analysis was performed on a PerkinElmer PE 2400 elemental analyzer. Melting points were determined on a Stuart SMP3 and are uncorrected. The monitoring of the reaction progress was performed by using TLC on Silufol UV254 plates. Heterocycles **5**, **6**, **7**, **8**, **11**, and **12** were synthesized in accordance with the literature’s data: **6**, **8** [14], **5**, **7** [25], **11**, and **12** [26]. All synthesized compounds are >95% pure by elemental analysis. Test compounds were dissolved in 99% DMSO (stock concentration 40 mM) and stored at −25 °C. If sediment or opalescence was detected, 5% *v*/*v* Tween 20 (Merck) was added. Serial dilutions were prepared ex tempore in a media suitable for the particular study. The final concentration in samples: DMSO < 0.25%, Tween 20 < 0.025% (were added to control samples in equal concentrations).

#### 4.1.1. Preparation of 7-Alkylamino-5-methyl-6-nitrotetrazolo[1,5-*a*]pyrimidines **9a**–**f** (General Method)

A solution of 7-chloro-5-methyl-6-nitrotetrazolo[1,5-*a*]pyrimidine (**7**) (1.1 g, 5.1 mmol) in anhydrous acetonitrile (20 mL) was treated first with triethylamine (0.71 mL, 5.1 mmol) and then with the appropriate alkylamine (5.1 mmol), maintaining the reaction mixture temperature in the range of 5–10 °C. After that, the reaction mixture was maintained at room temperature for 1 h and worked up according to the method described below.

#### 4.1.2. *N*-Isopropyl-5-methyl-6-nitrotetrazolo[1,5-*a*]pyrimidin-7-amine (**9a**)

The reaction mixture was evaporated to dryness at 30 °C. The product was purified by flash chromatography using EtOAc as the eluent. Yield 1.05 g (80%). Yellow crystals. Mp 53–55 °C. IR spectrum, *ν*, cm^−1^: 1329 (NO_2_), 1586 (NO_2_), 2134 (N_3_), 3345 (NH). ^1^H NMR spectrum (400 MHz, CDCl_3_), δ, ppm (*J*, Hz): **tautomer 9aA** (75%): 1.29 (6H, d, *J* = 8.0, 2CH_3_); 2.70 (3H, s, 5-CH_3_); 4.38–4.44 (1H, m, CH); 8.16–8.22 (1H, m, NH); **tautomer 9aB** (25%): 1.29 (6H, d, *J* = 8.0, 2CH_3_); 2.70 (3H, s, 5-CH_3_); 4.29–4.35 (1H, m, CH); 5.55–5.61 (1H, m, NH). ^13^C NMR spectrum (100 MHz, CDCl_3_), δ, ppm: 22.5 (2CH_3_); 26.1 (5-CH_3_); 43.8 (CH); 125.6 (C-6); 156.1 (C-7); 162.0 (C-3a); 168.9 (C-5). Found, %: C 40.82; H 4.44; N 41.33. C_8_H_11_N_7_O_2_. Calculated, %: C 40.51; H 4.67; N 41.33.

#### 4.1.3. *N*-*Tert*-Butyl-5-methyl-6-nitrotetrazolo[1,5-*a*]pyrimidin-7-amine (**9b**)

The reaction mixture was evaporated to dryness at 30 °C. The product was purified by flash chromatography (CHCl_3_). Yield 1.04 g (81%). Dark yellow crystals. Mp 49–51 °C. IR spectrum, *ν*, cm^−1^: 1333 (NO_2_), 1586 (NO_2_), 2130 (N_3_), 3364 (NH). ^1^H NMR spectrum (400 MHz, CDCl_3_), δ, ppm: **tautomer 9bA** (84%): 1.52 (9H, s, 3CH_3_); 2.67 (3H, s, 5-CH_3_); 8.35 (1H, s, NH); **tautomer 9bB** (16%): 1.52 (9H, s, 3CH_3_); 2.67 (3H, s, 5-CH_3_); 5.65 (1H, s, NH). ^13^C NMR spectrum (100 MHz, CDCl_3_), δ, ppm: 25.9 (5-CH_3_); 29.0 (3CH_3_); 53.9 (C); 126.3 (C-6); 156.5 (C-7); 161.5 (C-3a); 168.7 (C-5). Found, %: C 43.04; H 5.32; N 39.26. C_9_H_13_N_7_O_2_. Calculated, %: C 43.03; H 5.22; N 39.02.

#### 4.1.4. 2-[(5-Methyl-6-nitrotetrazolo[1,5-*a*]pyrimidin-7-yl)-amino]ethanol (**9c**)

The reaction mixture was evaporated to dryness at 30 °C. The product was purified by flash chromatography using CHCl_3_ as the eluent. Yield 0.85 g (70%). Yellow crystals. Mp 138–140 °C. IR spectrum, *ν*, cm^−1^: 1328 (NO_2_), 1592 (NO_2_), 2144 (N_3_), 3269 (OH), 3339 (NH). ^1^H NMR spectrum (400 MHz, CDCl_3_), δ, ppm (*J*, Hz): 2.49 (1H, br, OH); 2.70 (3H, s, CH_3_); 3.76 (2H, dt, *J* = 4.0, *J* = 12.0, NCH_2_); 3.87 (2H, t, *J* = 4.0, OCH_2_); 8.61 (1H, br, NH). ^13^C NMR spectrum (100 MHz, CDCl_3_), δ, ppm: 25.8 (CH_3_); 43.9 (NCH_2_); 61.1 (OCH_2_); 126.1 (C-6); 157.4 (C-7); 162.0 (C-3a); 168.8 (C-5). Found, %: C 35.13; H 3.77; N 41.00. C_7_H_9_N_7_O_3_. Calculated, %: C 35.15; H 3.79; N 40.99.

#### 4.1.5. 3-[(5-Methyl-6-nitrotetrazolo[1,5-*a*]pyrimidin-7-yl)-amino]propanol (**9d**)

The reaction mixture was evaporated to dryness at 30 °C. The product was purified by flash chromatography using EtOAc as the eluent and recrystallized from water. Yield 0.84 g (65%). Gray powder. Mp 86–88 °C. IR spectrum, *ν*, cm^−1^: 1325 (NO_2_), 1608 (NO_2_), 2130 (N_3_), 3271 (NH, OH). ^1^H NMR spectrum (400 MHz, CDCl_3_), δ, ppm: **tautomer 9dA** (73%): 1.86–1.92 (2H, m, CH_2_); 2.53 (1H, br, OH); 2.69 (3H, s, CH_3_); 3.73–3.79 (4H, m, NCH_2_, OCH_2_); 8.68 (1H, br, NH); **tautomer 9dB** (27%): 1.86–1.92 (2H, m, CH_2_); 2.53 (1H, m, OH); 2.70 (3H, s, CH_3_); 3.65–3.71 (4H, m, NCH_2_, OCH_2_); 6.91 (1H, br, NH). ^13^C NMR spectrum (100 MHz, CDCl_3_), δ, ppm: 25.9 (CH_3_); 31.4 (CH_2_); 39.3 (NCH_2_); 60.4 (OCH_2_); 125.8 (C-6); 157.1 (C-7); 162.0 (C-3a); 168.7 (C-5). Found, %: C 38.08; H 4.16; N 38.63. C_8_H_11_N_7_O_3_. Calculated, %: C 37.95; H 4.38; N 38.72.

#### 4.1.6. 3-[(5-Methyl-6-nitrotetrazolo[1,5-*a*]pyrimidin-7-yl)-amino]propane-1,2-diol (**9e**)

The reaction mixture was evaporated to dryness at 30 °C. The product was purified by flash chromatography using 1:1 CHCl_3_–EtOAc as the eluent and recrystallized from water. Yield 0.96 g (70%). Cream-colored powder. Mp 81–83 °C. IR spectrum, *ν*, cm^−1^: 1325 (NO_2_), 1606 (NO_2_), 2130 (N_3_), 3274 (NH), 3278 (OH). ^1^H NMR spectrum (400 MHz, DMSO-*d*_6_), δ, ppm(*J*, Hz): **tautomer 9eA** (72%): 1.72–1.78 (2H, m, CH_2_OH); 2.59 (3H, s, CH_3_); 3.49–3.62 (4H, m, HOCHCH_2_NH); 4.47 (1H, t, *J* = 4.0, HOCH_2_); 8.77–8.83 (1H, m, NH); **tautomer 9eB** (28%): 1.72 –1.78 (2H, m, CH_2_OH); 2.59 (3H, s, CH_3_); 3.49–3.62 (4H, m, HOCHCH_2_NH); 4.38 (1H, t, *J* = 4.0, HOCH_2_); 8.22–8.28 (1H, m, NH). ^13^C NMR spectrum (CDCl_3_), δ, ppm: 26.0 (CH_3_); 31.4 (CH); 39.3 (NCH_2_); 60.4 (OCH_2_); 125.8 (C-6); 157.1 (C-7); 162.0 (C-3a); 168.7 (C-5). Found, %: C 35.69; H 4.13; N 36.50. C_8_H_11_N_7_O_4_. Calculated, %: C 35.69; H 4.12; N 36.42.

#### 4.1.7. *N*-[2-(4-Chlorophenyl)ethyl]-5-methyl-6-nitrotetrazolo-[1,5-*a*]pyrimidin-7-amine (**9f**)

The obtained precipitate was filtered off and recrystallized from EtOH. Yield 1.53 g (90%). Yellow powder. Mp 150–151 °C. IR spectrum, *ν*, cm^−1^: 1087 (Cl), 1328 (NO_2_), 1585 (NO_2_), 2128 (N_3_), 3377 (NH). ^1^H NMR spectrum (400 MHz, DMSO-*d*_6_), δ, ppm (*J*, Hz): 2.58 (3H, s, CH_3_); 2.88 (2H, t, *J* = 7.2, NCH_2_CH_2_); 3.70 (2H, dt, *J* = 8.8, *J* = 6.0, NCH_2_); 7.20–7.26 (4H, m, H-Ar); 8.68 (1H, t, *J* = 5.2, NH). ^13^C NMR spectrum (100 MHz, DMSO-d6), δ, ppm: 24.1 (5-CH_3_); 33.6 (NCH_2_CH_2_); 42.2 (NCH_2_); 126.0 (C-6); 128.2 (C-2,6 Ar); 130.4 (C-3,5 Ar); 130.9 (CCl); 137.9 (C-1 Ar); 155.7 (C-7); 160.6 (C-3a); 166.1 (C-5). Found, %: C 46.66; H 3.62; N 29.31. C_13_H_12_ClN_7_O_2_. Calculated, %: C 46.79; H 3.62; N 29.38.

#### 4.1.8. *N*-[2-(4-Hydroxyphenyl)ethyl]-5-methyl-6-nitrotetrazolo-[1,5-*a*]pyrimidin-7-amine (**9g**)

A solution of 7-chloro-5-methyl-6-nitrotetrazolo[1,5-*a*]pyrimidine (**7**) (1.29 g, 0.006 mol) in DMF (10 mL) was stirred and treated by adding triethylamine (0.83 mL, 0.006 mol) followed by a solution of tyramine (0.82 g, 0.006 mol) in DMF (30 mL), while maintaining the reaction mixture temperature in the range of 4–9 °C. The reaction mixture was stirred for 24 h at room temperature and evaporated to dryness at 45 °C. The product was purified by flash chromatography using EtOAc as the eluent. Yield 1.42 g (75%). Red crystals. Mp 127–128 °C. IR spectrum, *ν*, cm^−1^: 1328 (NO_2_), 1586 (NO_2_), 2137 (N_3_), 3167 (OH), 3351 (NH). ^1^H NMR spectrum (400 MHz, DMSO-*d*_6_), δ, ppm (*J*, Hz): 2.52 (3H, s, CH_3_); 2.75 (2H, t, *J* = 7.2, NCH_2_CH_2_); 3.61 (2H, dt, *J* = 7.2, *J* = 6.4, NCH_2_); 6.67 (2H, d, *J* = 8.0, 2CH); 7.00 (2H, d, *J* = 8.0, 2CH); 8.70 (1H, t, *J* = 5.2, NH); 9.18 (1H, br, OH). ^13^C NMR spectrum (100 MHz, DMSO-*d*_6_), δ, ppm: 24.5 (5-CH_3_); 33.6 (NCH_2_CH_2_); 43.0 (NCH_2_); 115.2 (C-3,5 Ar); 126.0 (C-6); 128.9 (COH); 129.6 (C-2,6 Ar); 155.7 (C-7); 155.8 (C-1 Ar); 160.8 (C-3a); 166.4 (C-5). Found, %: C 49.52; H 4.14; N 30.88. C_13_H_13_N_7_O_3_. Calculated, %: C 49.52; H 4.16; N 31.10.

#### 4.1.9. Synthesis of 6-nitro-2-(prop-2-yn-1-ylsulfanyl)[1,2,4]triazolo[1,5-*a*]pyrimidin-7-amines **10f**, **10g** (General Method)

6-Nitro-2-(prop-2-yn-1-ylsulfanyl)[1,2,4]triazolo[1,5-*a*]pyrimidin-7-one (**8**) (0.50 g, 0.002 mol) and POCl_3_ (0.93 mL, 0.01 mol) were dissolved in MeCN (25 mL) in a round-bottom flask followed by the addition of pyridine (0.32 mL, 0.004 mol) at 70 °C. The resulting mixture was refluxed with stirring for 4 h, cooled to room temperature, and concentrated under reduced pressure at 35 °C. The residue was washed with cooled hexane (20 mL) and extracted with EtOAc (2 × 20 mL). The combined organic phase was added to a mixture of Et_3_N (0.42 mL, 0.003 mol in the case of compounds **10f**, or MeOH in the case of compound **10g**) at −3–0 °C. The resulting mixture was stirred for 1.5 h at 0–5 °C and worked up as described below for each product.

#### 4.1.10. *N*-(4-Chlorophenethyl)-6-nitro-2-(prop-2-yn-1-yl-sulfanyl)[1,2,4]triazolo[1,5-*a*]pyrimidin-7-amine (**10f**)

The precipitate (Et_3_N·HCl) was filtered off, the filtrate was washed with H_2_O (2 × 10 mL), dried over Na_2_SO_4_, and concentrated under reduced pressure at 30 °C. The residue was purified by flash column chromatography with EtOAc–CHCl_3_ 1:1 as the eluent. Yield 0.54 g (70%). Yellow powder. Mp 175–176 °C. IR spectrum, *ν*, cm^−1^: 637 (C≡CH), 809 (C–Cl), 1289 (NO_2_), 1580 (NO_2_), 3235 (NH), 3289 (C≡CH). ^1^H NMR spectrum (400 MHz, CDCl_3_), δ, ppm (*J*, Hz): 2.27 (1H, t, *J* = 2.4, C≡CH); 3.13 (2H, t, *J* = 7.2, CH_2_); 4.03 (2H, d, *J* = 2.4, SCH_2_); 4.67 (2H, dt, *J* = 7.2, *J* = 6.8, NCH_2_); 7.23 (2H, d, *J* = 8.4, H-Ar); 7.35 (2H, d, *J* = 8.4, H-Ar); 9.31 (1H, s, H-5); 9.67–9.73 (1H, br, NH). ^13^C NMR spectrum (100 MHz, DMSO-*d*_6_), δ, ppm: 19.0 (SCH_2_); 36.0 (CH_2_); 46.7 (NCH_2_); 75.9 (CH); 79.7 (C≡CH); 119.6 (C-6); 128.3 (C-2,6 Ar); 130.7 (C-3,5 Ar); 131.2 (C-4 Ar); 137.3 (C-1 Ar); 144.9 (C-7); 153.0 (C-5); 157.8 (C-3a); 165.6 (C-2). Found, %: C 49.51; H 3.30; N 21.48. C_16_H_13_ClN_6_O_2_S. Calculated, %: C 49.42; H 3.37; N 21.61.

#### 4.1.11. 4-{2-[(6-Nitro-2-(prop-2-yn-1-ylsulfanyl)[1,2,4]triazolo[1,5-*a*]pyrimidin-7-yl)amino]ethyl}phenol (**10g**)

The reaction mixture was concentrated under reduced pressure at 30 °C, H_2_O (30 mL) was added to the residue, and the precipitated product was filtered off and recrystallized from EtOH. Yield 0.43 g (58%). Yellow powder. Mp 214–215 °C. IR spectrum, *ν*, cm^−1^: 632 (C≡CH), 1215 (C–O), 1250 (NO_2_), 1581 (NO_2_), 3279 (C≡CH). ^1^H NMR spectrum (400 MHz, DMSO-*d*_6_), δ, ppm (*J*, Hz): 2.92–2.98 (3H, m, C≡CH, CH_2_); 4.03 (2H, d, *J* = 2.4, SCH_2_); 4.47 (2H, br, NCH_2_); 6.64 (2H, d, *J* = 8.4, H-Ar); 7.04 (2H, d, *J* = 8.4, H-Ar); 8.99 (1H, s, OH); 9.12 (1H, s, H-5); 9.88–9.94 (1H, br, NH). ^13^C NMR spectrum (100 MHz, DMSO-*d*_6_), δ, ppm: 19.0 (SCH_2_); 35.8 (CH_2_); 47.3 (NCH_2_); 74.0 (CH); 79.7 (C≡CH); 119.9 (C-6); 115.2 (C-3,5 Ar); 128.0 (C-4 Ar); 129.7 (C-2,6 Ar); 145.3 (C-7); 153.0 (C-5); 156.0 (C-1 Ar); 157.8 (C-3a); 165.6 (C-2). Found, %: C 52.01; H 3.86; N 22.55. C_16_H_14_N_6_O_3_S. Calculated, %: C 51.88; H 3.81; N 22.69.

#### 4.1.12. 2-(5-Nitrofur-2-yl)-5-methyl-6-nitro-1,2,4-triazolo[1,5-*a*]pyrimidin-7-one aminoguanidinium salt (**13**)

To a suspension of 0.500 g (0.00163 mol) 2-(5-nitronylfuran-2-yl)-5-methyl-6-nitro-1,2,4-triazolo[1,5-*a*]pyrimidin-7-one (**12**) in 5 mL water, a suspension of 0.222 g (0.00163 mol) of aminoguanidine bicarbonate in 5 mL of water was added. The resulting suspension was refluxed for 10 min and cooled to 25 °C. The solid product formed was collected by filtration and washed with H_2_O. Yield 0.49 g (85%). Gray powder. Mp > 300 °C. IR spectrum, *ν*, cm^−1^: 1314 (NO_2_), 1346 (NO_2_), 1640 (NO_2_), 1729 (C=O), 3341 (NH). ^1^H NMR spectrum (400 MHz, DMSO-*d*_6_), δ, ppm (*J*, Hz): 2.45 (3H, s, CH_3_), 4.68 (2H, s, NH_2_), 6.76 (2H, br s, NH_2_), 7.25 (2H, br s, NH_2_), 7.39 (1H, d, *J* = 3.8, H-3′), 7.82 (1H, d, *J* = 3.8, H-4′), 8.58 (1H, NH). ^13^C NMR spectrum (100 MHz, DMSO-*d*_6_), δ, ppm: 23.0 (CH_3_), 112.9 (C-3′), 114.2 (C-2′), 126.5 (C-5), 148.9 (C-1′), 150.8 (C-7), 151.6 (C-4′), 153.3 (C-3a), 156.6 (C-6), 158.5 (C-2), 158.8 (NH_2_-(C=NH_2_^+^)-NH-NH_2_). Found, %: C 34.56; H 3.25; N 36.76. C_11_H_12_N_10_O_6_. Calculated, %: C 34.64; H 3.15; N 36.75.

#### 4.1.13. 2-(5-Nitrofur-2-yl)-5-methyl-6-nitro-1,2,4-triazolo[1,5-*a*]pyrimidin-7-one guanidinium salt (**14**)

To a suspension of 0.500 g (0.00163 mol) 2-(5-nitronylfuran-2-yl)-5-methyl-6-nitro-1,2,4-triazolo[1,5-*a*]pyrimidin-7-one (**12**) in 5 mL water, 0.065 g (0.00163 mol) of NaOH was added. The resulting solution was stirred for 10 min at 25 °C. To the resulting mixture, 0.222 g (0.00163 mol) of guanidine hydrochloride in 5 mL of water was added. The reaction mixture was stirred for 10 min at 25 ⁰C. The solid product formed was collected by filtration and washed with H_2_O. Yield 0.46 g (83%). Gray powder. Mp 289–291 °C. IR spectrum, *ν*, cm^−1^: 1256 (NO_2_), 1539 (NO_2_), 1661 (C=O), 3358 (NH). ^1^H NMR spectrum (400 MHz, DMSO-*d*_6_), δ, ppm (*J*, Hz): 2.44 (3H, s, CH_3_), 7.00 (6H, br s, 2xNH_2,_ NH_2_^+^), 7.40 (1H, d, *J* = 3.9, H-3′), 7.83 (1H, d, *J* = 3.9, H-4′). ^13^C NMR spectrum (100 MHz, DMSO-d6), δ, ppm: 23.4 (CH_3_), 113.2 (C-3′), 114.6 (C-2′), 126.6 (C-5), 148.9 (C-1′), 151.0 (C-7), 151.8 (C-4′), 153.5 (C-3a), 156.7 (C-6), 157.9 (C-2), 159.0 (NH_2_-(C=NH_2_^+^)-NH_2_). Found, %: C 36.24; H 3.16; N 34.60. C_16_H_20_N_10_O_8_. Calculated, %: C 36.17; H 3.04; N 34.51.

### 4.2. Animals

All procedures with animals in the study were carried out under the generally accepted ethical standards for the manipulations of animals adopted by the European Convention for the Protection of Vertebrate Animals used for Experimental and Other Scientific Purposes (1986) and taking into account the International Recommendations of the European Convention for the Protection of Vertebrate Animals used for Experimental research (1997). All sections of this study adhere to the ARRIVE Guidelines for reporting animal research [48]. Male mice (21–24 g.) were housed 5 per cage in ambient lighting and 60% humidity. Animals had free access to water and food before the study.

### 4.3. Isolation and Treatment of Peritoneal Macrophages

Peritoneal macrophages (PM) were isolated from the peritoneal exudate of 30 male C57bl/6j mice. To accumulate PM, 1 mL of 3% peptone solution was injected intraperitoneally. After 3 days, the mice were euthanized by cervical dislocation. Cells of peritoneal exudate were obtained by aseptic washing of the abdominal cavity with 5 mL of sterile Hanks’s solution (+4–6 °C) without calcium and magnesium ions. The total number and viability of cells were assessed in a Goryaev counting chamber (Russia) with a 0.4% trypan blue staining (Sigma-Aldrich, St. Louis, MO, USA). The cell concentration was adjusted to 1.0 × 10^6^ cells/mL in DMEM (Gibco) supplemented with 2 mM L-glutamine (Gibco, Waltham, MA, USA), 10% heat-inactivated fetal bovine serum (BioClot, Bavaria, Germany), 100 U/mL penicillin, and 100 mg/mL streptomycin (Gibco) and plated 200 μL/well in 96-well transparent plates (SPL Life Sciences Co., Ltd., Pocheon-si, Korea). After 2 h at 37 °C in a humidified atmosphere with 5% CO_2_, the wells were washed to remove non-adherent cells. After 24 h of incubation, 20 μL of the supernatants were substituted with 20 μL of solutions of test compounds, followed by *E. coli* O127:B8 LPS (100 ng/mL final concentration) after 30 min. The experiments were run in 3 independent replicates.

### 4.4. Assay of Nitric Oxide (NO)

The accumulation of the nitrite anion (a stable end product of NO decomposition produced by iNOS) in supernatants was determined using a standard Griess reagent. Briefly, 50 μL of supernatants were collected 22 h after incubation of PM, and the test and control compounds were mixed with 50 μL of 1% sulfonamide in 2.5% H_3_PO_4_ and 50 μL of 0.1% *N*-(1-naphthyl) ethylenediamine in 2.5% H_3_PO_4_. After incubation at 23 °C for 10 min in an orbital shaker, the optical density was determined at a wavelength of 550 nm with a microplate reader Infinite M200 PRO (Tecan, Grödig, Austria).

### 4.5. Assay of Cytokines

The cell supernatant was collected and centrifuged at 1000 g for 20 min in a 2–16PK centrifuge (Sigma, Osterode am Harz, Germany). The concentrations of IL-6 and TNF-α were determined by ELISA using commercial kits (Cloud-clone, Houston, TX, USA) with a microplate reader Infinite M200 PRO (Tecan, Grödig, Austria).

### 4.6. Cytotoxicity Study

The activity of lactate dehydrogenase (LDH) in a cell culture medium served as a marker of membrane permeability and cell death. Aliquotes of supernatants were taken after 24 h of inoculation with test compounds, mixed with 250 μL of 0.194 nM NADH solution in 54 mM phosphate-buffered saline (pH 7.5). Then, 25 μL of a 6.48 mM pyruvate solution was added to the mixture. The optical density was followed at a wavelength of 340 nm for 20 min. The MTT-test was performed 24 h after incubations of cells with tested compounds. Briefly, 20 μL of MTT solution was added to each well, incubated at 37 °C in a humidified atmosphere containing 5% CO_2_ for 4 h. The culture medium was removed, the cells were lysed, and formazan crystals were dissolved in 150 μL DMSO. The plates were shaken at room temperature for 10 min, and the optical density was measured in a microplate reader Infinite M200 PRO (Tecan, Grödig, Austria) at a wavelength of 565 nm.

### 4.7. Phagocytosis Assay

Peritoneal macrophages of C57bl/6j mice were cultured in a 24-well plate at a volume of 500 μL/well (1 × 10^5^ cells/mL) for 24 h. Test compounds or DMSO were added to wells in triplicate followed by 50 μL of 1% yeast suspension, and the plates were incubated for 40 min. After 24 or 72 h, the medium was removed. The cells were fixed with May–Grunwald methanol dye and stained with Azur-Eosin in Romanovsky’s modification for 45 min at room temperature, washed, air-dried, and analyzed using a microscope Mikmed-6 (Russia), equipped with a digital camera. The study was performed in three technical replicates and two series; 100 macrophages were processed in each sample to count the captured yeast cells. Macrophage spreading was assessed as a number of pseudopodia.

### 4.8. LPS-Induced Acute Lung Injury

The randomization of C57BL/6J mice was performed by body weight and motor activity in an open field test. Dexamethasone (5 mg/kg) and **9g** (30 mg/kg) were administered intraperitoneally in 10 mL/kg sterile saline. The control animals received an equal volume of the vehicle. After 1 h, the mice were anesthetized with isoflurane inhalation until their breathing rate was decreased. The mice were suspended by the front incisors on an inclined surgical table, the tongue was pulled out with narrow curved tweezers, and 1 mg/mL of *E. coli* O127:B8 LPS (Sigma-Aldrich, St. Louis, MO, USA) in 1 mL/kg sterile saline was injected into the back of the oropharynx [49]. Intact control animals received an equal volume of sterile saline similarly.

### 4.9. Open Field Test

The animals were placed in the center of a round open field with an arena of 44 cm diameter and 32 cm wall height under 300 lux lighting. During 5 min of observation, horizontal motor activity, vertical motor activity, time spent in the central part, the number of exits to the central part, and exploratory activity parameters were recorded. Body temperature was determined rectally.

### 4.10. Bronchoalveolar Lavage and Plasma Preparation

The mice were anesthetized with 500 mg/kg chloral hydrate (Sigma-Aldrich, St. Louis, MO, USA) intraperitoneally 24 h after LPS administration. Blood was taken by intracardiac puncture in test tubes with heparin. The blood samples were centrifuged at 1000 g and 4 °C for 15 min on a 2–16PK centrifuge (Sigma, Germany), and plasma was separated and stored at −80 °C until the assay. Thoracotomy was performed, the ligature was applied to the left bronchus and the trachea was cannulated with a 20 G needle. The right lung was washed twice with 0.7 mL of warm sterile saline. After combining aliquots, the bronchoalveolar lavage (BAL) was centrifuged at 800 g and 4 °C for 10 min. The supernatant was separated and stored at −80 °C until the assay. The residual cell pellet was resuspended in 50 μL of PBS for further study. The left lung was washed in saline and placed in 10% buffered formalin for morphological evaluation.

### 4.11. Leukocyte Count in Blood and BAL

The total number of leukocytes in heparinized blood was determined after staining with methylene blue in Goryaev’s counting chamber at ×100 magnification. Smears of blood and BAL cell pellets were air-dried, fixed according to May–Grunwald for 3 min, and stained according to Romanowsky–Giemsa. After 30 min, the slides were washed, air-dried, and examined with immersion objective (×1000 total magnification). On each slide, a total of at least 100 cells were calculated.

### 4.12. Lung Permeability Index

The concentration of total protein in BAL supernatants was determined spectrophotometrically with the pyrogallol red method, and the protein in the blood plasma was determined with the biuret method using commercial kits (Vital, Saint Petersburg, Russia) and bovine serum albumin as the standard. The lung permeability index was calculated as the protein concentration in the BAL to plasma ratio.

### 4.13. Histological Study

Morphological markers of inflammation in lung tissue were assessed in a semi-quantitative way [33] on paraffin sections stained with hematoxylin and eosin. The sections were examined under a light microscope (Zeiss, Germany) by a double-blind method. The score of inflammation was determined as follows: 0—the presence of single inflammatory cells; 1—weak inflammation, inflammatory cells infiltrate no more than 10% of the lung tissue, including interalveolar cell partitions; 2—moderate inflammation, inflammatory cells infiltrate no more than 50% of the structures of the lung, but the interstitial tissue is identified; 3—severe inflammation, inflammatory cells densely infiltrate more than 50% of the lung tissue and airways. At the same time, lymphoid follicles localized around large and medium bronchi were not taken into account in the final assessment.

Paraffin sections of 5 μm thickness were mounted on slides treated with poly-*L*-lysine (Menzel GmbH & Co. KG, Braunschweig, Germany). After dewaxing and rehydration, they were incubated in 3% hydrogen peroxide for 20 min to block endogenous peroxidase. Immunostaining was carried out using MAX PRO (MULTI) peroxidase-polymer imaging system according to the manufacturer’s instructions (Histofine). The unmasking of antibodies was carried out by boiling sections at 100 °C in 0.01 M citrate buffer (pH 6.0) for 20 min. Sections of lung tissue were incubated with primary antibodies to CD68 (clone 3F-103, Santa Cruse Biotechnology, Dallas, TX, USA) at room temperature for 1 h and treated with 3,3′-diaminobenzidine. Finally, sections were stained with Mayer’s hematoxylin. The slides were studied and photographed using AxioScope.A1 microscope (Zeiss, Munich, Germany) equipped with an AxioCam MRc5 camera. The photos obtained were processed using ZENpro 2012 (Zeiss, Germany).

### 4.14. Data Analysis

Statistical analysis and graph preparation was performed in Prism 7.0 (GraphPad Software Inc., San Diego, CA USA). One-Way ANOVA with a Dunnett’s post-test was used for multiple comparisons and Mann–Whitney U-test for pairwise comparisons. IC_50_ values were calculated with nonlinear 3-parametric regression.

## Data Availability

Data is contained within the article.

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
