# Peer review of "Discovery of Nitro-azolo[1,5-*a*]pyrimidines with Anti-Inflammatory and Protective Activity against LPS-Induced Acute Lung Injury"

_pharmaceuticals, 2022, doi:10.3390/ph15050537_

Round 1

Reviewer 1 Report

In my opinion, the application of POCl3 for the real synthesis of pharmaceuticals have not any sens. Next, all synthetised compounds should be full characterised using spectral techniques. The question of the influence of nitro group on the bioactivity of target compounds should be discussed on the basis of data available in the literature.

Author Response

Comment

Response

In my opinion, the application of POCl3 for the real synthesis of pharmaceuticals have not any sens.

Indeed, POCl3 is a hazardous reagent. The authors agree with the reviewer's opinion that this is a serious limitation when using this reagent. However, highly effective purification methods are used for the target compounds, allowing to exclude the presence of harmful impurities in the studied substances, which is confirmed by the data of elemental analysis and other physico-chemical methods.

Next, all synthetised compounds should be full characterised using spectral techniques.

Thank you for pointing this out. We agree with this comment and there were done required corrections:

1. Chemical experimental part was added.

2. Synthesis procedures were added.

3. Supporting information file was added.

The question of the influence of nitro group on the bioactivity of target compounds should be discussed on the basis of data available in the literature.

We added some discussion about role of the nitro group in azoloazine heterocyclic family in the Introduction section.

Reviewer 2 Report

Comments to Author:

The authors synthesized 11 nitro pyrimidine compounds and tested the anti-inflammatory and LPS studies. In that, they identified one active compound (9g).

  1. I suggest to the authors add figure on the first page, helping the readers to understand the background of this work.
  2. The authors should need to mention/cited the synthetic procedure/references for the starting materials 6 and 11.
  3. In table 1 the compound codes mentioned as 6f, 6g, and 9. But those should be 10f,10g, and 13, please change accordingly.
  4. After seeing the results in table 1, compound 13 is also a promising compound. Did the authors check the further activity for compound 13? Or please comment on the author's ideas about that compound.
  5. The authors didn’t include the synthesis procedures, spectral data, and spectrums for all compounds, please make a supporting information file with all the information. So it will be helpful for the upcoming researchers.

Author Response

Comment

Response

I suggest to the authors add figure on the first page, helping the readers to understand the background of this work.

Thank you for the suggestion. Figure 2 and background details were added to the Introduction (lines 62-76).

The authors should need to mention/cited the synthetic procedure/references for the starting materials 6 and 11.

As suggested by the reviewer, we have added this information (line 340).

In table 1 the compound codes mentioned as 6f, 6g, and 9. But those should be 10f,10g, and 13, please change accordingly.

Thank you very much for the careful evaluation of our manuscript. Compound numbers have been corrected accordingly.

After seeing the results in table 1, compound 13 is also a promising compound. Did the authors check the further activity for compound 13? Or please comment on the author's ideas about that compound.

Thank you for pointing this out. Corresponding discussion has been added at lines 139-140.

The authors didn’t include the synthesis procedures, spectral data, and spectrums for all compounds, please make a supporting information file with all the information. So it will be helpful for the upcoming researchers.

Thank you for pointing this out. We agree with this comment and there were done required corrections:

1. Chemical part were added.

2. Synthesis procedures were added.

3. Supporting information file was added.

Reviewer 3 Report

The manuscript pharmaceuticals-1664402 devoted the actual field of organic and medicinal chemistry, namely new nitro-azolo[1,5-a]pyrimidines as anti-inflammatory and protective agents against LPS-induced acute lung injury and can be interested to the specialists working in this field. The authors’ opinion is clear and based on a good experimental material. The paper fit the Journal scope and formal requirements. However, it needs major revision before publication.

To improve the quality and perception of the manuscript I would suggest paying attention to following comments:

  1. Given that the authors have synthesized new compounds, only references to synthetic methods are insufficient. The authors need to discuss the synthesis of compounds, features of the spectra, and so on.
  2. In the context of the previous remark, it is necessary to add the experimental chemical part, as well as Supplementary information with copies of the spectra (1H and 13C NMR, LCMS, etc).
  3. There are some grammar and orthographical errors in the manuscript, which should be corrected

My decision is major revision.

Author Response

Comments

Response

Given that the authors have synthesized new compounds, only references to synthetic methods are insufficient. The authors need to discuss the synthesis of compounds, features of the spectra, and so on.

We added additional information about synthesis and properties of obtained heterocycles in the Results/Chemistry section.

In the context of the previous remark, it is necessary to add the experimental chemical part, as well as Supplementary information with copies of the spectra (1H and 13C NMR, LCMS, etc).

Thank you for pointing this out. We agree with this comment and there were done required corrections:

1. Chemical part were added.

2. Synthesis procedures were added.

3. Supporting information file was added.

There are some grammar and orthographical errors in the manuscript, which should be corrected.

English was checked throughout the manuscript and corrections were made.

Round 2

Reviewer 1 Report

Authors considered all my remarks and improved manuscript accordingly. In the present form, the paper can be further evaluated.

Reviewer 2 Report

The authors addressed all reviewer concerns. I suggest accepting the manuscript.

Reviewer 3 Report

The authors took into account the comments of the reviewer. The revised version may be accepted for publication.